# March 2019 dengue fever outbreak at the Kenyan south coast involving dengue virus serotype 3, genotypes III and V

Eric M. Muthanje[1,2], Gathii Kimita[2], Josphat Nyataya[2], Winrose Njue[2], Cyrus Mulili[2], Julius Mugweru[1], Beth Mutai[2], Sarah N. Kituyi[1], John Waitumbi[2]*

**1** Department of Biological Sciences, University of Embu, Embu, Kenya, **2** United States Army Medical Research Directorate-Africa, Basic Science Laboratory, Kisumu, Kenya

* john.waitumbi@usamru-k.org

**Data Availability Statement:** Sequence data for this study are publicly available in the GenBank database under the accession numbers

## Abstract

The first description of a disease resembling dengue fever (DF) was in the 15th century slave trade era by Spanish sailors visiting the Tanzania coast. The disease, then associated with evil spirits is now known to be caused by four serotypes of dengue virus (DENV1-4) that are transmitted by *Aedes* mosquitoes. Kenya has experienced multiple outbreaks, mostly associated with DENV-2. In this study, plasma samples obtained from 37 febrile patients during a DF outbreak at Kenya's south coast in March 2019 were screened for DENV. Total RNA was extracted and screened for the alpha- and *flavi*-viruses by real-time polymerase chain reaction (qPCR). DENV-3 was the only virus detected. Shotgun metagenomics and targeted sequencing were used to obtain DENV whole genomes and the complete envelope genes (E gene) respectively. Sequences were used to infer phylogenies and time-scaled genealogies. Following Maximum likelihood and Bayesian phylogenetic analysis, two DENV-3 genotypes (III, n = 15 and V, n = 2) were found. We determined that the two geno-types had been in circulation since 2015, and that both had been introduced independently. Genotype III's origin was estimated to have been from Pakistan. Although the origin of geno-type V could not be ascertained due to rarity of these sequences globally, it was most related to a 2006 Brazilian isolate. Unlike genotype III that has been described in East and West Africa multiple times, this was the second description of genotype V in Kenya. Of note, there was marked amino acid variances in the E gene between study samples and the Thailand DENV-3 strain used in the approved Dengvaxia vaccine. It remains to be seen whether these variances negatively impact the efficacy of the Dengvaxia or future vaccines.

## Introduction

Dengue fever (DF), believed to be a corruption of Kiswahili words "*Ka-dinga pepo*" signifying a disease characterized by a sudden cramp-like seizure, caused by an evil spirit was first described in 15th century by Spanish sailors visiting the then Tanganyika (Tanzanian today) coast [1]. Despite these early reports, there exist sparse data on the extent of its transmission,

[MZ544494 - MZ544507] and [MZ544585 - MZ544588]

**Funding:** JW, P0136_19_KY by Armed Forces Health Surveillance Branch (AFHSB) and it's GEIS (Global Emerging Infections Surveillance and Response) Section. URL: https://health.mil/Military-Health-Topics/Combat-Support/Armed-Forces-Health-Surveillance-Division/Global-Emerging-Infections-Surveillance-and-Response. The funders had no role in study design, data collection and analysis, decision to publish, or preparation of the manuscript.

**Competing interests:** The authors have declared that no competing interests exist.

and impact in Africa. This can be attributed to limited surveillance and unavailability of affordable diagnostics that can differentiate DF from other "look alike" tropical diseases that are endemic in sub-Saharan Africa. Globally, dengue virus (DENV) is estimated to cause 3.9 billion infections and between 20,000 to 70,000 fatalities annually [2]. In Africa, 20% of the population is estimated to be at risk of DENV infections, with 1 million cases of symptomatic DF occurring each year [3].

DENV belongs to the *Flaviviridae* family and is transmitted by *Aedes aegypti* and *Ae. albopictus* mosquitoes, which are distributed in tropical regions and are highly adapted to human habitats [4]. DENV is a positive sense RNA virus whose genome is approximately 11 kb, with three structural proteins (capsid, pre-membrane, and envelope) and seven non-structural proteins (NS1, NS2A, NS2B, NS3, NS4A, NS4B, and NS5) [5–7]. The envelope (E) protein is a target for humoral immunity and therefore a good vaccine candidate [8]. Dengvaxia developed by Sanofi-Pasteur, is the only approved DF vaccine and is derived from the pre-membrane and envelope proteins. There are major concerns on the use of Dengvaxia due to its limited efficacy and possible harms in DENV-seronegative hosts who later acquire a natural DENV infection [9].

DENV has four known virus types (DENV-1-4), referred to as serotypes, that are antigenically and genetically distinct [6] and all of them have been reported in Africa, with DENV-2 and DENV-1 causing majority of the epidemics [10, 11]. Evolutionary studies show that each serotype evolved independently from their common ancestor in a series of divergence events [12]. The serotypes share 65–70% amino acid similarity [13] and are further subdivided into genotypes based on their genetic diversity and sometimes geographical distribution [12, 14, 15]. For instance, DENV-1 and DENV-3 have five genotypes (I-V), DENV-2 has six, and DENV-4 has four genotypes [12, 16, 17]. The genotypes of DENV-3 vary by approximately 3% at the amino acid level and typically do not exceed 6% nucleotide divergence [12, 18].

Infection with one DENV serotype does not cross-protect across other serotypes. In fact, remnant sub-neutralizing antibodies from previous infections with different serotypes are considered a major risk factor for the development of the severe, life threatening dengue hemorrhagic fever/dengue shock syndrome [19]. The pathophysiology of the severe dengue is mostly associated with antibody dependent enhancement (ADE) due to interaction of remnant sub-neutralizing antibodies and DENV surface proteins that allow increased infection of Fc-receptor bearing host cells [19, 20].

The first recorded case of DF in Kenya was attributed to DENV-2 and was identified in 1982 in a Canadian tourist who had visited Malindi, Kilifi County at the Kenyan coast [21]. The same serotype was again isolated in Kilifi in 1997 [22]. Several outbreaks were later reported in the period between 2011–2014 in Mandera and Mombasa Counties in North Eastern Kenya and at the Kenyan coast respectively and were attributed to DENV-1-3 [23, 24]. Between 2014–2015, the cosmopolitan genotype of DENV-2 was detected in Kilifi [25]. Two years later, the same genotype of DENV-2 was again reported in an outbreak that occurred in 2017 in Malindi [26]. In the same year, DENV-2 was reported in North Eastern Kenya and at the Kenyan coast [2]. DENV-2 of the cosmopolitan lineage dominated in most of these outbreaks [26–28]. The first case of DENV-4 was later reported in Western Kenya [29].

Clearly, the coastline of Kenya dominates in DF outbreaks, a finding that is backed by seroprevalence endemicity studies. For instance, Malindi, an old town along the Kenyan coast had an IgG seroprevalence of 34.17%, while Busia in western Kenya had a seroprevalence of 1.96% [30]. What drives this disparity is not clearly understood, considering that competent mosquito vectors are abundant in the two regions [31]. Part of this disparity can be attributed to limited surveillance and limited availability of specific diagnostics for dengue, as well as lack of

expertise and awareness of the disease by the clinicians [32]. More recently, DENV-1-4 were detected in a 2014–2015 pediatric febrile surveillance study in Western Kenya [33, 34].

In the past decade, there has been increased reports of DF outbreaks at the Kenyan coast probably due to improved diagnosis, surveillance, and enhanced spread of the virus as a result of the ever increasing trans human movement from endemic to non-endemic areas. Collectively, the Viral Pathogen Resource (VIPR) [35] contained over 24,295 DENV genome sequences as at March 2021, of which only 262 are from Africa (DENV-1 = 47, DENV-2 = 98, DENV-3 = 74 and DENV-4 = 13). Of these, Kenya has contributed 51 sequences (DENV-1 = 5, DENV-2 = 30, DENV-3 = 8 and DENV-4 = 8). To redress the surveillance gap and under-representation of Africa DENV sequences, the US Army has invested heavily in DENV genomic surveillance. A recent genomic study on Kenya's DENV-2 obtained during a 2017 DF outbreak in Malindi indicated trends toward the emergence of new immunogens that are relevant to vaccine design [28]. In an effort to continue to remedy the scarcity of Kenyan DENV genome sequences, we report on DENV-3 sequences obtained from a 2019 DF outbreak at the Kenyan south coast.

## Methods

### Study design

37 plasma samples were collected during the March 2019 DF outbreak from febrile patients seeking hospital care at Mtongwe Naval Base Hospital, Mombasa County. The samples were collected under an ongoing acute febrile illness surveillance protocol that was reviewed and approved by Kenya Medical Research Institute's Scientific and Ethical Research Unit (KEMRI SERU #1282) and the Walter Reed Army Research Institute's Human Subject Protection Branch (WRAIR #1402). A written informed consent was obtained from each participant.

### RNA extraction, DENV serotyping and sequencing

Total RNA was extracted from the plasma samples using Genetic Signature's virus nucleic acids extraction kit (Genetic Signatures, NSW, Australia) on the MagPurix Evo automated extraction System (Zinexts Life Science Corp, Taiwan). An aliquot of the extracted RNA was screened for the presence of *alpha-* and *flavi*-viruses using the EasyScreen typing kit (Genetic Signatures, NSW, Australia) according to the manufacturer's instructions and as previously described [36].

RNA for sequencing was extracted using the Direct-zol miniprep kit (Zymo Research, CA, USA). To enrich for viral pathogens, the host nucleic acids were depleted by treatment with Ribo-Zero Gold rRNA removal kit (Illumina, CA, USA). cDNA synthesis was performed by sequence-independent single-primer amplification (SISPA) [37], followed by cDNA amplification using MyTaq DNA polymerase (Bioline, MA, USA). Sequence libraries were prepared using Nextera XT kit (Illumina, CA, USA) and a 12 pM library was then sequenced using MiSeq reagent v.3 600 cycles (Illumina, CA, USA) on MiSeq platform (Illumina, CA, USA).

To obtain complete E genes from the specimens with partial DENV sequences, cDNA from these samples was amplified using E gene specific primers. Briefly, the complete DENV-3 E gene (~1,479 bp) was amplified in a nested PCR using primers that were designed in house using NCBI's Primer-Blast tool [38]. For the primary amplification, the forward DENV-3-610E: `CGACAAGAGATCAGTGGCGT` and reverse (DENV-3-2999E: `GCCATGTGCAGGTTTT-CACC` primers were used. For the subsequent nested amplification, a second primer pair comprising DENV-3-824E: `AAGGTCTGTCAGGAGCTACG` as forward primer and DENV-3-2410E: `CTTCCACCTCCCACACATTCC` as reverse primer were used. The PCR reactions contained 2 μL of the cDNA, 2 μL of 0.5 μM primer sets and 1 μL of 5 U of MyTaq DNA polymerase

(Bioline, MA, USA). Thermal cycling conditions for the first and second PCR were similar and included initial denaturation at 95˚C for 1 min, 35 cycles of denaturation at 95˚C for 1 min, annealing at 56˚C for 30 sec, extension at 72˚C for 1 min and final extension at 72˚C for 5 min. Non-target controls (PCR water) were used to track contamination. Amplified products were visualized on 1% agarose gels stained with GelRed (Biotium Inc., USA). Amplicons were cleaned using AmpureXP beads (Beckman Coulter, CA USA) and used to make sequence libraries with the Nextera XT kit (Illumina, CA, USA). 12 pM of the library was sequenced on the MiSeq platform, with 600 v.3 paired end chemistry (Illumina, CA, USA).

## Sequence assembly and phylogenetics

Demultiplexed sequence reads from both shotgun RNA sequencing and targeted sequencing were retrieved from the Miseq. The reads were processed using an in-house pipeline. In brief, the raw reads were passed through Trimmomatic v 0.22 to remove short sequences (less than 40 nucleotides), remove sequencing adapters, failed reads and reads with poor base quality scores (<Q30). The reads were then filtered using cutadapt v 1.1 to clip surviving adapters and SISPA primers. For the shotgun assay, the clean reads were mapped against the host (human genome reference—Hg38) using bowtie v2 to subtract reads matching to the human genome. The host depleted reads were then assembled *de novo* using Ray v2.3. The generated contigs were taxonomically identified by querying them against other DENV sequences using BLASTn in the non-redundant nucleotide database (GenBank). The DENV genome whose homology was closest to our sequences (GenBank accession: MK894339.1) was retrieved and used for a reference guided assembly in the CLC Genomics workbench v 8.5.1 (QIAGEN, CA, USA).

The sequences from the targeted (E gene sequencing) and shotgun assays (whole genome) were mapped against the MK894339.1 reference genome using CLC Genomics workbench v 8.5.1 with default parameters. To call the consensus sequence, a minimum base depth coverage of 5 and the majority consensus rule was employed. In case of positions with conflicting base calls, quality scores were used to resolve conflicts.

To determine the phylogenetic relationships between the Kenyan DENV-3, a comprehensive subset of curated, annotated and published DENV-3 dataset was obtained from VIPR [35]. The datasets were down sampled by geographical location, year of collection, genotype and presence of the full genomes, or the presence of complete E genes (see S2 Table). In brief, VIPR database was accessed on 14th March 2021 to retrieve context sequences for phylogeny. Only complete sequences with a known collection date, genotype, and geographical location were used. Sequences were clustered and cluster representatives selected ensuring consideration of the indicated set criteria. Where multiple cluster members shared similar temporal-spatial information, representatives were randomly selected. These global datasets and the Kenyan samples were used to conduct two multiple sequence alignments with Muscle v 3.8 [39], the first involving the complete polyprotein coding sequence, and another involving the entire E protein. Alignments were manually edited using CLC Genomics workbench v8.5.1 in order to fix any misalignments that might have been inadvertently introduced by the alignment algorithm. To avoid using sequences from recombination of different virus strains during sequence assembly, the aligned sequences were run in RDP4 software suite using RDP GENECONV and Bootscan [40]. Sequences deemed to have recombinants were excluded from downstream analysis. Maximum likelihood (ML) trees were inferred in PhyML v3.1 [41] using the Generalized time-reversible (GTR) substitution model with gamma distribution and invariant sites (GTR+Γ+I) determined by AIC and BIC tests in jModelTest v2. Branch support was estimated by approximate likelihood ratio test values [42].

### Time-scaled genealogies

A strict molecular clock with coalescent constant size was used for molecular time clock analysis of the E gene. The suitability of the E gene tree for use in the estimation of temporal parameters was first evaluated in TempEst software v1.5.3 [43]. Root-to-tip regression indicated adequate temporal signal (correlation coefficient = 0.6 for n = 398 sequences). Time-scaled phylogeny was then inferred using BEAST v 1.10 package [44]. The GTR+Γ+I model of substitution estimated by jModelTest2 [42] was used to estimate the time tree. 200 Million MCMC chains were performed, and convergence of runs were estimated in Tracer v1.7.1 [43] ensuring all effective sample sizes were >200. The maximum-clade credibility tree was summed with Tree Annotator and visualized in FigTree v1.4.3.

### Homology estimates of the DENV-3 E protein to the Thailand DENV-3 strain used in the approved Dengvaxia vaccine

CLC Genomics Main workbench v8.5 was used to create alignments and compute genetic distances between the 17 DENV-3 E genes from this study and to the DENV-3 strain from Thailand DENV-3 (PaH881/8 (AF349753) that is used in the approved Dengvaxia vaccine [45]. The amino acid comparison was performed by aligning the study sequences with the Thailand strain PaH881/88 sequence available in the GenBank. Comparisons were also made to other African DENV-3 sequences available in the GenBank, and this information is shown in S1 Table.

## Results

### The 2019 dengue fever outbreak was caused by DENV-3

Only DENV-3 was identified at screening and was present in 21 out of the 37 specimens examined. Only 4/21 samples yielded complete genomes. The rest (17/21) yielded partial genomes, of which only 3/17 had the complete E gene. The remaining 14 samples were sequenced using targeted amplicon approach and 10 yielded complete E gene. Four samples failed to produce E gene on targeted sequencing. Table 1 shows the relationship between the qPCR cycle threshold (Ct) values and the subsequent success of obtaining whole genomes from the sequencing effort. By Shotgun metagenomics, four specimens with Ct values between 20 and 28 generated complete DENV-3. Three specimens with Ct values between 27 and 32 generated partial DENV-3 genomes but with the complete E gene. Ten specimens with Ct values between 25 and 36 generated partial DENV-3 genomes with incomplete E gene. Complete E gene was obtained after targeted sequencing of the 10 samples. Four samples with Ct values >36 failed to produce E gene on targeted sequencing.

### Phylogenetic tree derived from the complete genome of Kenya DENV-3

Whole genomes from the 4 Kenya DENV-3 samples and those sampled from the global dataset (n = 221) were used for phylogenetic analysis. The closest polyprotein gene homologues (99.73% nucleotide identity) was with a DENV-3 of genotype III that was identified in a Chinese traveler who had visited Tanzania (Accession number MK894339.1). The sequences also shared a most recent common ancestor (MRCA) with the Asian DENV-3 from Pakistan.

### DENV-3 genotype III and V were co-circulating during the 2019 DF outbreak at the Kenyan coast

DENV-3 E genes from this study (n = 17) and those sampled from the global dataset (n = 383) were used to infer phylogenetic trees. Both the ML and Bayesian phylogenetic trees were

**Table 1. The relationship between the qPCR cycle threshold (Ct) values and the subsequent success of sequencing.** Four specimens with Ct values between 20 and 28 generated complete DENV-3 by Shotgun metagenomics. Three specimens with Ct values between 27 and 32 generated partial DENV-3 genomes but with the complete E gene. Ten specimens with Ct values between 25 and 36 generated partial DENV-3 genomes with incomplete E gene. Complete E gene was obtained after targeted sequencing of these 10 samples. Four samples with Ct values >36 failed to produce E gene on targeted sequencing.

| Sample ID | Ct values at screening | WGS reads after QC | Reads mapped to DENV-3 | Genome size | Comment |
|---|---|---|---|---|---|
| MTW-341 | 28 | 2,551,431 | 1,798,595 | 10,173 | **Complete DENV-3 on WGS** |
| MTW-359 | 20 | 2,521,271 | 2,340,939 | 10,173 | |
| MTW-355 | 28 | 2,036,567 | 1,459,058 | 10,176 | |
| MTW-4167 | 25 | 5,336,058 | 3,738,625 | 10,173 | |
| MTW-34985 | 28 | 2,428,899 | 388,665 | 1,479 | **Partial DENV-3 with complete E gene on WGS** |
| MTW-419 | 32 | 3,254,873 | 33,518 | 1,479 | |
| MTW-361 | 27 | 298,825 | 199,869 | 1,479 | |
| | | **E gene sequence reads after QC** | **Reads mapped to DENV-3** | **E gene size** | |
| MTW-364 | 27 | 262,527 | 196,252 | 1,479 | **Complete E gene after targeted sequencing** |
| MTW-338 | 28 | 758,802 | 527,412 | 1,479 | |
| MTW-3158 | 30 | 803,246 | 539,219 | 1,479 | |
| MTW-4157 | 29 | 256,334 | 231,405 | 1,468 | |
| MTW-348 | 32 | 326,746 | 292,989 | 1,479 | |
| MTW-34748 | 27 | 6,904 | 2,904 | 1,479 | |
| MTW-344 | 25 | 265,065 | 237,275 | 1,479 | |
| MTW-339 | 35 | 294,683 | 259,984 | 1,479 | |
| MTW-362 | 35 | 399,162 | 362,503 | 1,442 | |
| MTW-337 | 36 | 176,713 | 157,288 | 1,439 | |
| MTW-342 | 39 | 2,333 | 78 | 468 | **No DENV sequences** |
| MTW-347 | 38 | 4,999 | 16 | 1059 | |
| MTW-365 | 38 | 4,170 | 8 | 175 | |
| MTW-346 | 37 | 76 | 0 | 0 | |

concordant. As shown in Fig 2A–2C), majority of the samples (n = 15, 88.2%) branched with the genotype III (Fig 2B) in a well-supported monophyletic clade (aLRT and Posterior probability = 1). The others (n = 2, 11.8%) branched with genotype V (Fig 2C). Genotype III (Fig 2B) branched together with a sequence from China (Genbank accession number MK894339). This clade shared a MCRA with Asian genotype III from Pakistan and India. Genotype III E sequences had a nucleotide sequence similarity > 99.3% (S1 Fig). The other 2 Kenyan samples (Fig 2C) branched in a monophyletic clade with genotype V sequences collected between 1963–2006 from the USA, Japan, China, Brazil and Philippines. This clade formed a sister clade with representatives of four other Kenyan sequences obtained in Western Kenya between the year 2014–2015. The genotype V samples had nucleotide sequence similarity of 99.6% (S1 Fig).

## DENV-3 from the 2019 Coastal Kenya outbreak has been in circulation since 2015

The molecular clock analysis was used to estimate the evolutionary rates of the dataset (n = 398 sequences) under a strict coalescent constant size with the least likelihood (-18826.8). The substitution rate of DENV-3 was estimated to be 7.7E-4 (95% Highest Posterior Density HPD: 6.9–8.4). As shown in Fig 3, the molecular clock analysis indicated that the TMRCA for the Kenyan DENV-3 genotype III was in 2015 (95% (HPD): 2013–2016, node probability = 1)

**Table 2. Sequence alignment of the 17 Kenyan DENV-3 E proteins to the DENV-3 vaccine strain from Thailand (AF349753).** The first 15 amino acid sequences are genotype III and the last 2 are genotype V.

| Amino acid Position | 10 | 13 | 14 | 81 | 93 | 124 | 147 | 154 | 169 | 217 | 270 | 271 | 275 | 301 | 340 | 377 | 383 | 391 | 426 | 447 | 452 | 478 | 489 | Genotypes |
|---|---|---|---|---|---|---|---|---|---|---|---|---|---|---|---|---|---|---|---|---|---|---|---|---|
| AF349753 | R | E | G | I | K | S | N | D | V | P | N | S | G | L | G | V | K | K | V | S | I | V | A | III |
| MTW-341 | - | - | - | V | - | P | D | E | T | - | - | - | - | T | E | I | N | - | - | - | V | I | - | |
| MTW-419 | - | - | - | V | - | P | D | E | T | - | - | - | - | T | E | I | N | - | - | - | V | I | - | |
| MTW-34985 | - | - | - | V | - | P | D | E | T | - | - | - | - | T | E | I | N | - | - | - | V | I | - | |
| MTW-348 | - | - | - | V | - | P | D | E | T | - | - | - | - | T | E | I | N | - | - | - | V | I | - | |
| MTW-339 | - | - | - | V | - | P | D | E | T | - | - | - | - | T | E | I | N | - | - | - | V | I | - | |
| MTW-359 | - | - | - | V | - | P | D | E | T | - | - | - | - | T | E | I | N | - | - | - | V | I | - | |
| MTW-361 | - | - | - | V | - | P | D | E | T | - | - | - | - | T | E | I | N | - | - | - | V | I | - | |
| MTW-355 | - | - | - | V | - | P | D | E | T | - | - | - | - | T | E | I | N | - | - | - | V | I | - | |
| MTW-4167 | - | - | - | V | - | P | D | E | T | - | - | - | - | T | E | I | N | - | - | - | V | I | - | |
| MTW-338 | - | - | - | V | - | P | D | E | T | S | - | - | - | T | E | I | N | - | - | - | V | I | - | |
| MTW-34748 | - | - | - | V | - | P | D | E | T | - | - | - | - | T | E | I | N | - | A | - | V | I | - | |
| MTW-344 | T | - | S | V | - | P | D | E | T | - | - | - | - | T | E | I | N | - | - | - | V | I | - | |
| MTW-4157 | - | G | - | V | - | P | D | E | T | - | - | - | - | T | E | I | N | - | - | - | V | I | - | |
| MTW-362 | - | - | - | V | - | P | D | E | T | - | - | - | - | T | E | I | N | - | - | - | V | I | - | |
| MTW-337 | - | - | - | V | - | P | D | E | T | - | - | - | - | T | E | I | N | - | - | - | V | I | - | |
| MTW-3158 | - | - | - | - | E | - | D | E | A | - | T | L | I | - | - | - | - | R | - | G | - | I | V | V |
| MTW-364 | - | - | - | - | E | - | D | E | A | - | T | L | - | - | - | - | - | R | - | - | - | I | V | |

Note: The dash indicates conserved residues.

from Pakistan, suggesting dispersal between Asia and East Africa. The same analyses estimated the TMRCA for the Kenyan genotype V to have existed in 2015 (95%HPD: 2012–2017, node probability = 1). These two sequences branched with a sequence from Brazil collected in 2006. The exact origins of the Kenyan genotype V from this study remain unclear, and the long branch leading to the Kenyan genotype V suggests limited local and/or global sequence data for this genotype. This lack of sufficient data is also shown in previously reported genotype V from Kenya [29].

### Kenya's DENV-3 differ significantly from the Thailand DENV-3 strain used in the approved Dengvaxia vaccine

Table 2 shows the amino acids comparison between the 17 DENV-3 E proteins from this study and the PaH881/8 (AF349753) strain from Thailand that is incorporated in the currently approved tetravalent, live attenuated Dengvaxia vaccine [46]. There are 23 amino acid variances between study strains and the vaccine strain. Similar variances were observed in the other African DENV-3 sequences available in the GenBank (S1 Table).

### Discussion

DF outbreaks at the Kenyan coast have recently become an annual occurrence [2, 24–28, 34, 47]. The cause of this increased transmission is probably multi-factorial, including the rapid movement of people from one region to the other, increase in population growth, increased urbanization [48] and probably changes in weather patterns that may be favoring increased activities of *Ae. Aegypti* and *Ae. albopictus*, the mosquito vectors responsible for transmitting DENV within countries neighboring the Indian Ocean [49].

Most cases of DF occur during the rainy season that results in accumulation of water pools which provide breeding habitats for mosquitoes [47]. The hot and humid tropical climate at the Kenyan coast receives bimodal rainfall with long rains experienced between March and June and short rains from August to October [50]. In the current study, we sought to characterize DENV-3 strains associated with a DF outbreak that occurred in March 2019 at south coast of Kenya.

Unlike in the previous DF outbreaks that were caused by DENV-2 of the cosmopolitan lineage [25–28], the March 2019 outbreak was caused by DENV-3. The last time DENV-3 was reported in Kenya was in 2011–2014 in Mandera, and these outbreaks are believed to have originated from Somalia [23, 24]. Before that, DENV-3 had been reported in 1984 and 1985 at the coastal region of the Indian Ocean, specifically Pemba and Mozambique [51, 52]. It was later detected between 1992 and 1993 in the United States troops serving in Somalia, and also in the areas neighboring the Persian Gulf [53]. In other parts of Africa, DENV-3 has been reported in West Africa (Senegal, Côte d'Ivoire, Togo, Benin) and East Africa (Djibouti, Somalia, Tanzania, Madagascar) and phylogenetic studies trace these outbreaks to the Indian subcontinent [54–56].

Advanced characterization of DENV relies on WGS. As shown in Table 1, our attempts to obtain complete genomes were only successful in 4 out of 21 specimens that had tested positive for DENV-3 by qPCR. Using qPCR Ct values as surrogate for viral load, we evaluated whether the failure to obtain WGS following shotgun metagenomics was related viral load, as has been suggested by Thorburn et al., [57] and Cruz et al., [58]. As shown in Table 1, the Ct values of the 4 samples with complete genomes (>10,000 bp) ranged from 20 to 28. Other specimens with similar Ct values produced partial genomes or needed targeted sequencing in order to generate the complete E gene. It was noted that samples producing complete sequences had over 1,459,000 reads mapping to DENV-3. Samples that produced incomplete genomes had much lower sequence reads ($\leq$539,000) while samples with Ct values $\geq$37 did not generate DENV sequences even after targeted sequencing for E gene. To better quantify the inverse relationship of Ct values and recovery of complete genomes, future studies would benefit from the use of a control/s with known copy numbers. We speculate that the failure to obtain WGS when Ct values were within "sequenceable" range was due to factors such as sample integrity and/or presence of background genomes from the host or commensal microbiome, as has been alluded to in a previous reports [59].

Four DENV-3 with complete polyprotein-coding sequences (10,173–10,176 bp) were assembled. The coding sequence lies between nucleotides positions 81 and 10,250, and encodes a polyprotein of 3,392 amino acids. As shown in Fig 1, the closest polyprotein gene homologues to the 4 sequences with 99.73% nucleotide identity was with a DENV-3, genotype III that was identified in a Chinese traveler (GenBank accession number MK894339.1) who had visited Tanzania in 2018. Tanzania has experienced several DF outbreaks in the last decade, the most recent having occurred in 2018 and was associated with DENV-1 and DENV-3 [60]. It is likely that the outbreak reported by Chipwaza et al., [60] could have seeded the 2019 DF outbreak at the Kenyan coast.

Whole genome sequencing is considered the gold standard for DENV systematics [17], but because there more partial genomes in global repositories than full genomes, genes such as NS1, NS3 and NS5 and E genes are also used. Despite the fact that NS1, NS3 and NS5 exhibit significantly higher phylogenetic resolution than E-gene [17], the latter has been the choice of many phylogenetic studies mostly because of its historical role in diagnostics, its influence on viral infectivity and immunogenicity [61–64]. The latter consideration is crucial for evaluating potential mismatch of the currently licensed E gene based (Dengvaxia) to circulating DENV strains. Importantly, E gene sequences are over represented in public repositories. For

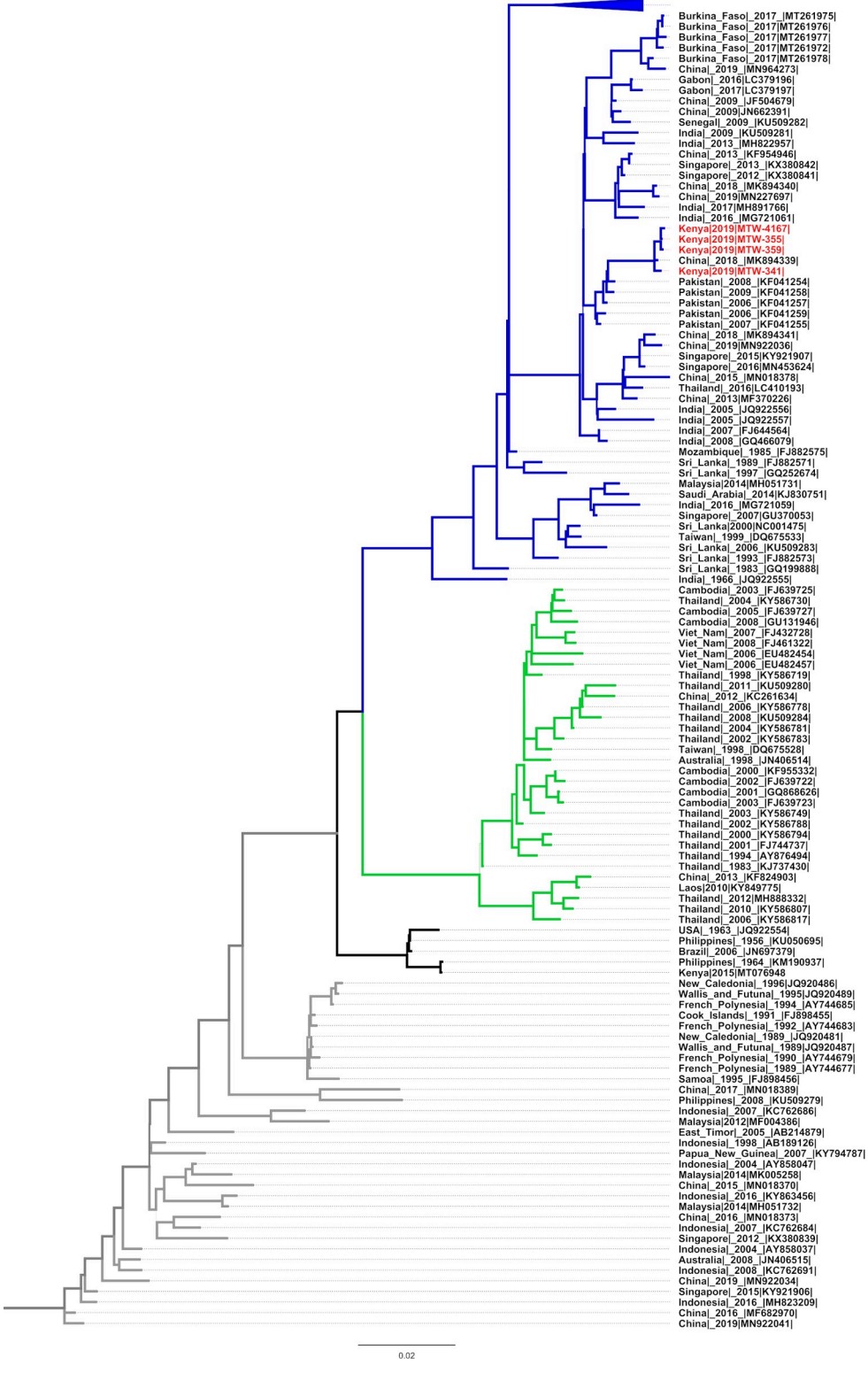

**Fig 1. DENV3 phylogenetic tree including the 4 Kenyan DENV-3 complete polyprotein sequences.** Kenyan DENV-3 are indicated in red fonts, while global representatives are coded by genotype type: Blue = Genotype III, Grey = Genotype I, Black = Genotype V and Green = Genotype II. The scale bar represents genetic distance. Thin lines indicate posterior probability values of <1.

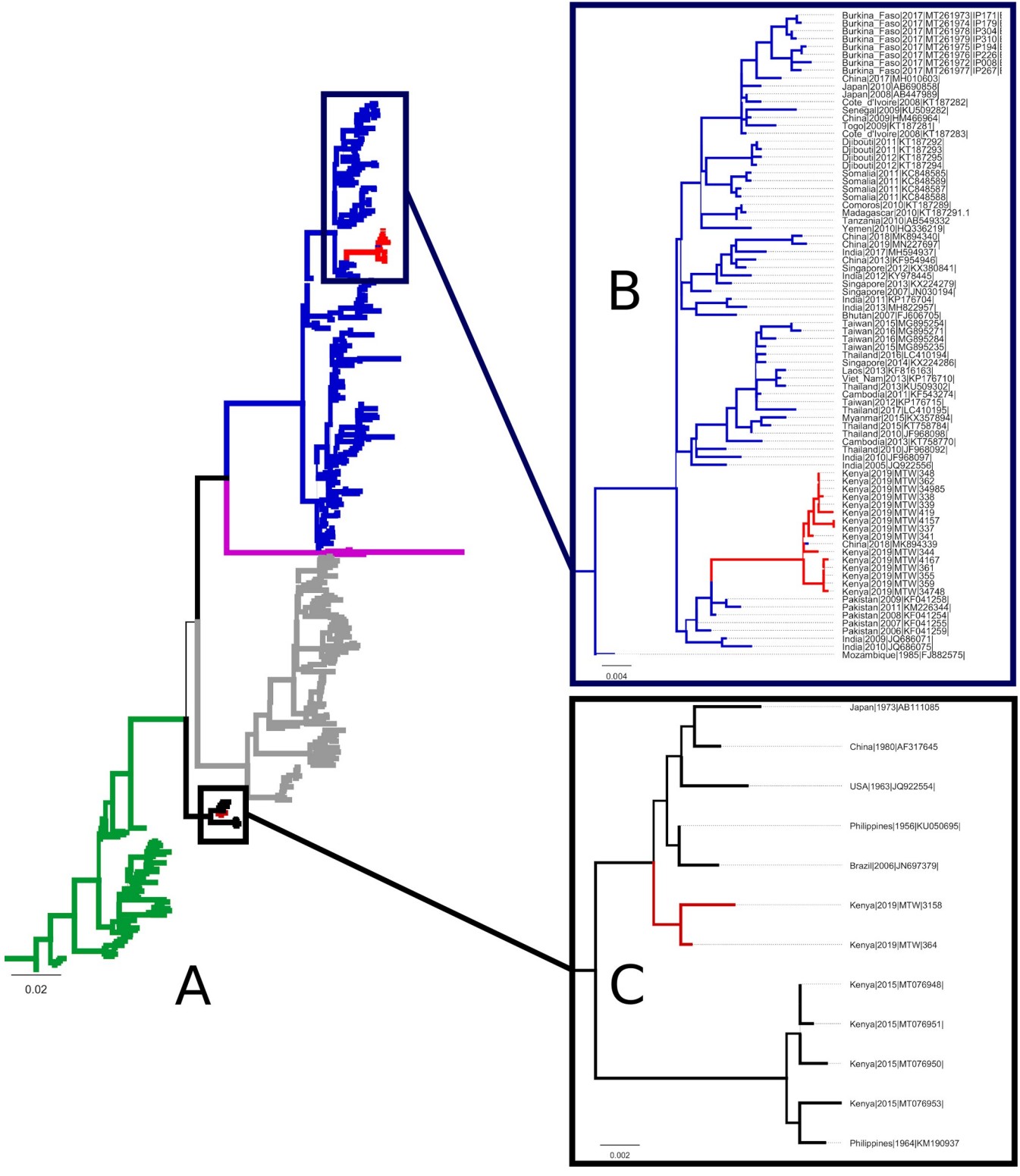

**Fig 2.** A. A summarized maximum likelihood phylogeny showing the sequence analysis of Kenyan samples from this study and DENV-3 strains retrieved from the NCBI GenBank. Fig 2B: A branch section of genotype III showing clearly the branching of the Kenyan samples with other derived global sequences. Fig 2C: Two Kenyan genotype V clade branching in a monophyletic clade with genotype V sequences. Samples are color coded by genotype. Blue = Genotype III, Magenta = Genotype IV, Grey = Genotype I, Black = Genotype V and Green = Genotype II. The scale bar represents genetic distance. Thin lines indicate posterior probability values of < 1.

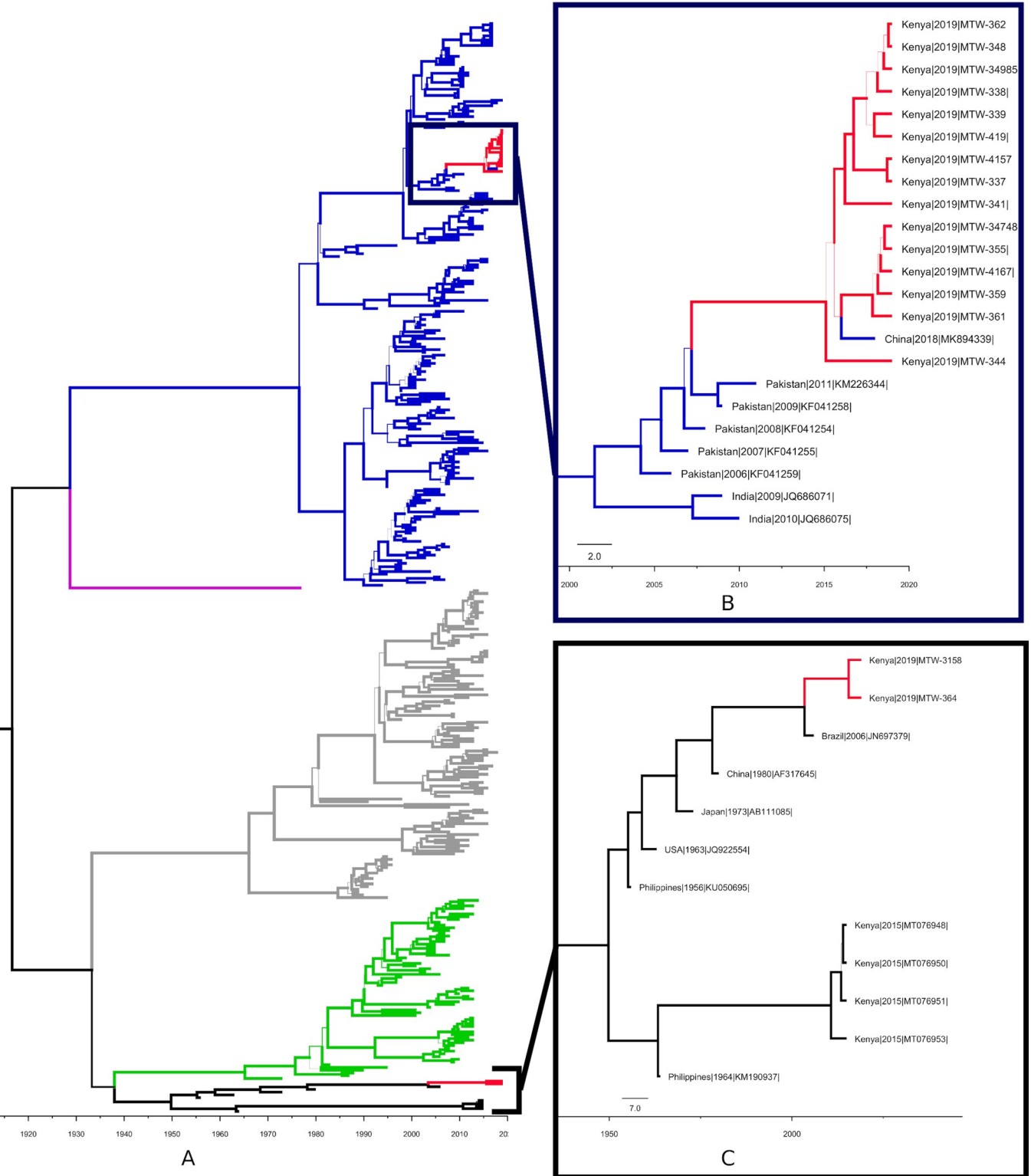

**Fig 3.** Bayesian time-scaled phylogeny of the DENV-3 E sequences from this study with a global representative subset (Fig 3A). Fig 3B shows a blow out of the time tree with a subset of genotype III isolates (Blue lines) branching with genotype III samples from this study (Red lines). Fig 3C shows a blow out of the time tree showing the genotype V clade. In this clade, Kenyan genotype V samples are shown as Red lines, while global isolates as black lines. Across all the trees, Blue = Genotype III, Magenta = Genotype IV, Grey = Genotype I, Black = Genotype V and Green = Genotype II. The scale represents time in years. Thin lines indicate posterior probability values of <1.

instance, globally for DENV-3, there are 4,678 E gene sequences compared to 1,420 full genomes, 2,003 NS1, 1,810 NS3, and 1,977 NS5 as of 4th November 2021 [35]. Africa DENV-3 sequences are far fewer in the VIPR repository, but still, E-gene sequences predominate: 89 compared to 11 full genomes, 35 NS1, 31 NS3 and 33 NS5. Thus, phylogenies constructed with E gene are more robust. For these reasons, the E genes from the 17 samples (4 from samples with complete DENV-3 genomes, 3 from samples with complete E gene following shotgun metagenomics, and 10 with full E gene following targeted sequencing, Table 1) were used to infer phylogenetic relationships. As shown in Fig 2A, of the 17 samples, 15 clustered with genotype III (Fig 2B) and two with genotype V (Fig 2C). This is not the first report of the two dengue genotypes co-circulating in the same outbreak. A similar occurrence was reported in Guangzhou China in 2009 where DENV-3 genotype III and V were in circulation during the same period [65].

Fig 3, shows the Bayesian time-scaled phylogeny of the DENV-3 E sequences from this study together with global representative subset. As shown in Fig 3B, which is an outcrop from Fig 3A, all the Kenyan genotype III sequences branched with those from Pakistan, suggesting a high likelihood that the strains share genealogy with those from Pakistan. Pakistan has experienced multiple DF outbreaks over the past two decades [66]. The most recent occurred in Peshawar, Capital city of Khyber Pakhtunkhra, Pakistan in 2017 where DENV-2 and DENV-3 were dominant [67]. It has been suggested that, the Pakistan DENV strains trace their evolutionary history to the East African coast [1]. The tree topology of the genotype V clade (Fig 3C) has two lineages with their TMRCA dating back to 1949 (95%HPD: 1945–1953). Lineage one contained two Kenyan south coast samples from this study (MTW-3158 and MTW-364) while lineage two contained four previously reported genotype V that were collected in 2015 from Western Kenya [29] (MT076948, MT076951, MT076950 and MT076953). It is tempting to speculate that two divergent genotype V lineages are in circulation in Kenya, probably geographically split into a coastal and Western Kenya population. We are cognizant of the limitation of this speculation, especially due to under sampling of DENV-3 genotype V in the country as well as globally. Currently, apart from the Brazil (2006) and Kenyan samples (2015 and 2019), the clade is populated by old viruses, the most recent being from China (1980). Members of the genotype V lineage are rarely reported and have previously even been considered extinct [68].

Time-scaled genealogies indicate that the DENV-3 genotype III reported in the current study have been in circulation since 2015 (Fig 3B). Considering sequences from this 2019 outbreak branched with a 2018 sequence from a Chinese traveler returning from Tanzania, there is a high probability that DENV-3 genotype III is endemic in East Africa though these conclusions suffer from the limitation of under sampled DENV-3 genomes in East Africa. Genotype V is also estimated to have been in circulation in 2015. However due to the rarity of genotype V sequences globally, it is difficult to clarify the genealogical time-scaled trees in this clade.

There are only 89 African DENV-3 sequences in the GenBank as at 04 November 2021 [35]. Our study adds 17 more DENV-3 sequences. As shown in Table 2, there are 23 amino acid variances between DENV-3 E sequences in the study samples and the DENV-3 strain from Thailand that is a component of the tetravalent, live attenuated Dengvaxia vaccine [46]. Similar variances were observed in the other African DENV-3 sequences available in the GenBank (S1 Table). It is not clear how much impact such variances will have in the efficacy of the current Dengvaxia vaccine or future vaccines. As a limitation, it is important to point out that the observed E protein variances did not include the sequences encoding the DENV pre-membrane (prM) gene, which together with the E protein constitute the tetravalent Dengvaxia vaccine.

## Conclusion

DENV-3 strains of genotypes III and V were identified to have been circulating locally years before the outbreak, with their TMRCA dating back to 2015. Our study suggests that DENV-3 circulating at the Kenyan coast share genealogy with those from Asia, especially Pakistan and this could have been facilitated by tourism and trade. The marked amino acid variances between the study samples and the Thailand DENV-3 strain used in the approved Dengvaxia vaccine may help inform future design of dengue vaccines. The fact that these variances, albeit from a small sample set, were similar to those in other African DENV-3 sequences and different from the vaccine representative strain from Thailand are indicative of the need to obtain more DENV sequences from African countries.

## Supporting information

**S1 Fig. Pairwise comparison of Dengue 3 E-gene sequences from our study samples.** The 15 genotype III sequences (shown in orange) had a nucleotide sequence similarity > 99.3%. The other 2 genotype V samples (shown in blue) had nucleotide sequence similarity of 99.6% (S1 Fig). The genotype III and V sequences differed by 6.7%.
(TIFF)

**S1 Table. Amino acid comparison of the envelop protein of DENV-3 sequences from Africa, available in the GenBank, against the parental DENV-3 strain from Thailand used as a component of the Dengvaxia vaccine.**
(DOCX)

**S2 Table. Dengue isolates used in this study for the E-gene phylogenetic analysis.** The datasets were down sampled by geographical location, year of collection, genotype and presence of the full genomes, or the presence of complete E genes.
(DOCX)

## Acknowledgments

We are grateful for the research subjects who participated in this study. The funders had no role in study design, data collection and analysis, decision to publish, or preparation of the manuscript.

## Disclaimer

Material has been reviewed by the Walter Reed Army Institute of Research. There is no objection to its publication. The opinions or assertions contained herein are the private views of the author, and they are not to be construed as official, or as reflecting true views of the Department of the Army or the Department of Defense. The investigators have adhered to the policies for protection of human subjects as prescribed in AR 70–25.

## Author Contributions

**Conceptualization:** John Waitumbi.

**Data curation:** Eric M. Muthanje, Josphat Nyataya, Winrose Njue, Cyrus Mulili, Beth Mutai.

**Formal analysis:** Eric M. Muthanje, Gathii Kimita, Sarah N. Kituyi.

**Funding acquisition:** John Waitumbi.

**Investigation:** Josphat Nyataya, Winrose Njue, Cyrus Mulili, Julius Mugweru, Sarah N. Kituyi, John Waitumbi.

**Methodology:** Eric M. Muthanje, Gathii Kimita, Josphat Nyataya.

**Project administration:** Beth Mutai.

**Resources:** John Waitumbi.

**Software:** Gathii Kimita.

**Supervision:** Julius Mugweru, Sarah N. Kituyi, John Waitumbi.

**Validation:** Gathii Kimita.

**Visualization:** Eric M. Muthanje, Gathii Kimita.

**Writing – original draft:** Eric M. Muthanje.

**Writing – review & editing:** Gathii Kimita, Julius Mugweru, Sarah N. Kituyi, John Waitumbi.

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
