## [Decision Letter · Decision Letter 0]

28 Sep 2021

PGPH-D-21-00281

Dengue fever outbreak at the Kenyan south coast involving serotype 3, genotypes III and V

Dear Dr. Waitumbi,

Thank you for submitting your manuscript to PLOS Global Public Health. After careful consideration, we feel that it has merit but does not fully meet PLOS Global Public Health’s publication criteria as it currently stands. Therefore, we invite you to submit a revised version of the manuscript that addresses the points raised during the review process.

We look forward to receiving your revised manuscript.

Kind regards,

Louisa Alexandra Messenger, MSc, PhD

Academic Editor

Journal Requirements:

1. We noticed that you used “data not shown”/"unpublished data" in the manuscript. We do not allow these references, as the PLOS data access policy requires that all data be either published with the manuscript or made available in a publicly accessible database. Please either remove these references, or amend the supplementary material to include the referenced data.

2. We do not publish any copyright or trademark symbols that usually accompany proprietary names, eg (R), (C), or TM  (e.g. next to drug or reagent names). Therefore please remove all instances of trademark/copyright symbols throughout the text, including MagPurix Evo® on page 5 and Dengvaxia® on reference 42 and 43.

3. In the online submission form, you indicated that your data will be submitted to a repository upon acceptance.  We strongly recommend all authors deposit their data before acceptance, as the process can be lengthy and hold up publication timelines. Please note that, though access restrictions are acceptable now, your entire data will need to be made freely accessible if your manuscript is accepted for publication. This policy applies to all data except where public deposition would breach compliance with the protocol approved by your research ethics board. If you are unable to adhere to our open data policy, please kindly revise your statement to explain your reasoning and we will seek the editor's input on an exemption. Please be assured that, once you have provided your new statement, the assessment of your exemption will not hold up the peer review process.

Additional Editor Comments (if provided):

The manuscript by Muthanje et al. reports the use of genomic data to evaluate a Dengue Fever (DF) outbreak in Kenya’s south coast in March 2019. Recurrent epidemics have been reported from the coastal areas of Kenya (Agha et al. 2019), with the latest warning having issues as recently as May of 2021 (https://reliefweb.int/report/kenya/kenya-dengue-fever-outbreak-emergency-plan-action-epoa-n-mdrke048). Genomic data from previous outbreaks (2011 and 2017) has been previously analysed and reported of Dengue Virus (DENV) serotype 2 (DENV-II) in the same region (Langat et al. 2020), with a particular interest in the co-circulation of two different Cosmopolitan genotype lineages (DENV-II genotype Cosmopolitan, also named DENV-II genotype II). Unlike the previously reported studies, this study reports the causative agent of the outbreak as DENV serotype 3 (DENV-III), belonging to the genotypes III and V. Unlike serotype classification, which differs from each other by 25–40 % at the amino acid level and can be easily detected through the qPCR targeting the targets the Capsid (C) and pre-Membrane (preM) region, the genotyping is obtained though full genome homology, vary by up to 3 % at the amino acid level (Diamond and Pierson 2015). High-throughput sequencing (HTS) technologies allow the simultaneous identification and characterization by serotyping and genotyping of DENV cases at the nucleotide level in a single methodological step, with previous studies having demonstrated that complete viral sequences can be obtained directly from patient sera using a shotgun metagenomics approach (Yozwiak et al. 2012). Therefore, the methodology used in this study has been successfully used in similar studies, with success in inferring the genetic relationship between DENV strains.

Major Comments

1. The authors started with a collection of 37 plasma samples from febrile patients, of which 21 were confirmed through qPCR positive for DENV-3 (56.75%). This is a low number of DENV samples, and no context is provided of the timeframe and geographical distribution of these samples. The only information provided was they were collected in March of 2019 from the Kenyan south coast, but no indication if they were from the same hospital or medical care centre was provided.

2. Of the 21 DENV-3 positive samples, the complete genomes were obtained only in 4 samples (19.04%). The authors describe the workflow used in the CLC Genomics Workbench, following the standard procedure of read filtering based on quality and de novo assembly. No information on the thresholds used for quality filtering is indicated, nor the average quality score of the remaining sequences. The authors also describe the use of reference guided assembly, without indicating its purpose. It is therefore unclear if the assemblies are obtained through de novo assembly or through guided assembly, and the caveats of consensus-based assembly are also not indicated. In previous studies, full DENV genomes were assembled even in samples containing as low as 960 DENV reads (Mendes et al. 2020).

3. The CT values of the 21 DENV-3 positive samples ranged between 20 and 39. According to previous studies with the typing kit used, the pan-alphavirus assay had a sensitivity range of 10–25 copies per reaction depending on the viral strain (Garae et al. 2020). The authors relate the high CT values with the unsuccess of the assembly of the viral genomes. It is important to include controls to clearly associate viral load with viral sequences obtained, and the success of the assembly.

4. The Kenyan DENV-3 dataset comprising the full DENV genomes assembled (n=4) was supplemented with 221 sequences from VIPR, filtered by geographical location, year of collection, genotype and presence of the full envelope protein. It is unclear why the query was not limited to full sequences, as genotyping is performed on the homology of the full coding sequence. Additionally, no indication of the values for the filters was made available, which strongly hinders the replication of the search query. A full list of accession numbers should also be made available as supplemental material.

5. It is stated by the authors that the multiple sequence alignments of the complete DENV coding sequences and envelope protein region were manually edited using CLC Genomics Workbench, no information is stated on the purpose or type of edition that was performed.

6. The conclusions drawn from the co-circulation of genotype III and V are drawn from the send of 17 envelope sequences obtained from the original 21 Kenyan sample dataset. This includes the 4 assembled complete genomes. The global dataset obtained from VIPR includes 383 sequences, which is assumed that also are partial sequences. Given that only envelopes sequences are used, the genotype classification is greatly hindered as other genetic regions, such as NS1, NS3, and NS5 have been shown to exhibit higher phylogenetic support (Cuypers et al. 2018). Therefore, whole-genome sequences provide superior classification precision and are particularly relevant when assessing the emergence and co-occurrence of DENV strains belonging to the same serotype, but different genotype. This is also relevant in the molecular clock analysis performed, where the 398 envelope sequences were used to estimate the evolutionary rate of the dataset. Limiting this analysis to a region and not the full genomic sequence of the DENV greatly impacts the results obtained.

7. Whole-genome targeted sequencing of DENV samples has been successfully used to obtain complete sequences from clinical samples (Parameswaran et al. 2017) and is a viable and cost-effective option for obtaining reliable data for phylogenetic inference and should be preferred to methods selecting a particular region of the genome.

8. DENGVAXIA vaccine contains chimeric yellow fever dengue viruses with serotype 1 to 4. It is constructed using recombinant DNA technology by replacing the sequences encoding the pre-membrane (prM) and envelope (E) proteins in yellow fever(YF) 17D204 vaccine virus genome with those encoding for the homologous sequences of dengue virus serotypes 1, 2, 3, and 4, respectively. The analysis performed only contained the DENV genomic region coding for the envelope. The pre-membrane region should not be excluded when analysing the potential loss of efficacy of the vaccine in the Kenyan DENV-3 samples.

9. Supplemental material should be made available for peer review as it might contain or might be missing important information for the comprehension of the manuscript.

Minor Comments

1. In accordance with the previous comments, the title should be changed to “Dengue fever outbreak at the Kenyan south coast involving serotype 3”.

2. Support should be added to the phylogenetic trees (Figure 1, Figure 2, and Figure 3)

3. In the Introduction, a few DENV serotypes and genotypes are stated by name but no clear description of the DENV classification, both at the serotype and genotype level, is presented.

4. An appropriate reference should be added to support the statement in the introduction that “The serotype share 65-70% aminoacid similarity”.

5. In the introduction, there’s a missing space between “DENV-1-3” and the reference [19, 20].

6. In the introduction, the following phrases are overly complicated and very confusing to read: “This could be due to an increase in [the] use of DF diagnosis, and/or DF surveillance. It could also be due to an increase in DENV in the population due to increased vector-human interaction and/or importation driven by an increase in transhuman movements from endemic countries”.

7. In the methods section, Homology estimates of the DENV-3 env protein to the Thailand DENV-3 strain used in the approved Dengvaxia vaccine, it is stated that “comparisons were also made to other African DENV-3 sequences available in GenBank”. The accessions of the sequences used should be stated.

8. A small typo is present in the second phase of the “DENV-3 from the 2019 Coastal Kenya outbreak has been in circulation since 2015” section, where ‘.,’ are present.

Reviewers' comments:

Reviewer's Responses to Questions

**Comments to the Author**

1. Does this manuscript meet PLOS Global Public Health’s publication criteria? Is the manuscript technically sound, and do the data support the conclusions? The manuscript must describe methodologically and ethically rigorous research with conclusions that are appropriately drawn based on the data presented.

Reviewer #1: Yes

Reviewer #2: Yes

2. Has the statistical analysis been performed appropriately and rigorously?

Reviewer #1: N/A

Reviewer #2: Yes

3. Have the authors made all data underlying the findings in their manuscript fully available (please refer to the Data Availability Statement at the start of the manuscript PDF file)?

Reviewer #1: Yes

Reviewer #2: Yes

4. Is the manuscript presented in an intelligible fashion and written in standard English?

Reviewer #1: Yes

Reviewer #2: No

5. Review Comments to the Author

Reviewer #1: The subject of the manuscript, the observation of dengue virus serotype 3 causing the outbreak at the Kenyan south coast is an important notification, as like authors also state in the manuscript, the recent outbreaks in Kenya have typically been caused by DENV-2. The article is well written and pleasant to read. This article provides important information about DENV-3 strains circulating in Kenya. There are however, issues that should be addressed in my opinion.

Specific comments

Title

I would indicate the year of the outbreak already in the title

Abstract

The number of patients would be nice to be stated already in the abstract

Line 19 please change RNA was isolated to RNA was extracted

Introduction

Line 42 Flavivirus is genus, Flaviviridae family

Line 42 please correct Ae. albopictus

Line 43 Ae. aegypti and Ae. albopictus are not globally distributed. The distribution area of especially Ae. aegypti is limited to tropical and subtropical regions. Please correct.

Line 46 Authors state that DENV E gene is more conserved. This is now a desultory remark as the authors do not explain more conserved compared to what?

Line 57, remove for instance, that is not needed.

Line 72-73 The sentence about pathophysiology of dengue virus infection needs to be rephrased. Even though ADE is strongly associated with severe cases of DENV infections, there are cases of severe dengue also with primary infections.

Line 80 Please add reference to Aedes aegypti abundance at the coast and in Busia

In general: Please describe the outbreak, when where, how many cases, reference

Materials and methods

Line 100 please rephrase “The study used 37 plasma samples”.

In Study design please tell abit more about the patient material. Were the patients diagnosed with dengue virus infection? The sample size is quite small so were there other inclusion criteria than just suspected dengue virus infection? What kind of disease/symptoms did they have? The hospital the samples were from is written in the manuscript but it would be good also to describe the cities/municipalities the patients were from.

Line 121 it is more common to talk about E gene than env gene.

In general few points about Materials and methods. The authors should describe how they have confirmed the validity of the results. For example data of positive controls in PCR and how they have excluded the possibility of contamination.

Discussion

Line 269 please change DF infection to dengue virus infection

Lines 295-297 “Judging from this data, it is clear that, apart from viral load, other factors such as sample integrity and/or presence of other contaminating genomes are crucial for the success WGS.” The authors should explain what they mean with contaminating genomes regarding to their sample materials/analyses.

I would like the discussion to include more analyses of the relation of the sequences produced in this study and previous DENV-3 sequences from Kenya.

Figures and tables

Figure 2 Please mark genotypes to the figure. The large figure’s resolution is not good.

General comments

Based on the results it might be quite strong statement to say that the virus is of Pakistani origin. It might be good to discuss the Asian origin in more general.

Comparison of the Kenyan sequences to the Dengvaxia is a bit irrelevant in my opinion. It is not stated which genotype the DENV-3 used in the vaccine is and it is obious that there are differences if the genotype is not the same. And as it is stated in the manuscript, the differences are the same as with previous virus strains from Kenya. To my knowledge, the vaccine is not in use in Kenya. So, I would leave this comparison and discussion out of the article.

Reviewer #2: Line 1: The Kenyan south coast is not defined. Where does it start and where does it end? If it is not properly defined, I recommend to the authors to mention specific locations where the study was undertaken in the title. I would also propose to refer to the serotypes and genotypes as being those of dengue virus (DENV).

Line 4: Affiliation #1 is missing the city, while this is indicated in affiliation #2.

Line 9: Why is superscript #2 is indicated along with the corresponding author’s name?

Line 14: Reconsider your reference to slave trade. Is it worthy mentioning and reminding the reader of this?

Line 18: I recommend including sample size. All of a sudden the numbers start in the results in line 23. Refer to line 177.

Line 20: Define all abbreviations at first mention throughout the text. qPCR has. not been defined at first mention.

Line 21: “whole genomes” of what?

Line 33:”corruption” is a very strong word. Consider an alternate word.

Line 35: Was it Tanzania or Tanganyika at that time?

Line 42: Does dengue virus belong to “Flavivirus”? Kindly indicate an appropriate family.

Line 50: What is dengue infection? Do you mean dengue virus infection?

Line 53: Please read and cite et al., Viruses. 2021 Mar 24;13(4):536. doi: 10.3390/v13040536.

Line 63, line 68: Add space between the ending of a sentence and the references.

Line 72: You have previously used DF for dengue. Ensure consistency.

Line 88: What about the role of imported tyres.

Line 100: What is the demography of study participants? How were the subjects selected? What was the case definition used? Was there status for other febrile illnesses checked?

Line 108: Include location (city) for the manufacturer. Do this throughout the text.

Line 114: Please check whether abbreviations at the beginning of the sentence is allowed.

Line 119: italicize gene names.

Line 124: Describe how nested PCR approach was used?

Line 140:is it blastn not supposed to be italicized?

Line 145: Define abbreviation VIPR

Line 176: Refrain from concluding statements in the titles. Correct this throughout.

Line 178: why use short cuts like 4/21?

Line 183 – 188: Discuss the ct value inconsistency in the Discussion. Follow guidelines for formatting Tables.

Line 245: There is a repeated word.

Line 295: Consider adding references to your discussion.

Line 337: Can this be proven from the outbreak samples during 2015?

6. PLOS authors have the option to publish the peer review history of their article (what does this mean?). If published, this will include your full peer review and any attached files.

**Do you want your identity to be public for this peer review?** For information about this choice, including consent withdrawal, please see our Privacy Policy.

Reviewer #1: No

Reviewer #2: **Yes: **Gerald Misinzo

---

## [Editor Report · Decision Letter 1]

9 Jan 2022

Dengue fever outbreak at the Kenyan south coast involving serotype 3, genotypes III and V

PGPH-D-21-00281R1

Dear Dr. Waitumbi,

We're pleased to inform you that your manuscript has been judged scientifically suitable for publication and will be formally accepted for publication once it meets all outstanding technical requirements.

Within one week, you'll receive an e-mail detailing the required amendments. When these have been addressed, you'll receive a formal acceptance letter and your manuscript will be scheduled for publication.

An invoice for payment will follow shortly after the formal acceptance. To ensure an efficient process, please log into Editorial Manager at https://www.editorialmanager.com/pgph/ click the 'Update My Information' link at the top of the page, and double check that your user information is up-to-date. If you have any billing related questions, please contact our Author Billing department directly at authorbilling@plos.org.

Kind regards,

Louisa Alexandra Messenger, MSc, PhD

Academic Editor